# Skill Neologisms: Towards Skill-based Continual Learning

Antonin Berthon [1]   Nicolas Astorga [1]   Mihaela van der Schaar [1]

## Abstract

Modern LLMs show mastery over an ever-growing range of skills, as well as the ability to compose them flexibly. However, extending model capabilities to new skills in a scalable manner is an open problem: fine-tuning and parameter-efficient variants risk catastrophic forgetting, while context-based approaches have limited expressiveness and are constrained by the model's effective context. We explore *skill neologisms*–soft tokens integrated in the model's vocabulary and optimized to improve capabilities over a specific skill–as a way to selectively acquire new skills without weight updates. We first observe that pre-trained LLMs already exhibit tokens associated with procedural knowledge. We then show on a controlled synthetic task that skill neologisms can be learned to improve model capabilities on specific skills while being composable with out-of-distribution skills, and that independently trained skill neologisms can be composed zero-shot. Finally, we validate zero-shot composition of independently learned skill neologisms on the more realistic natural language setting of the Skill-Mix benchmark (Yu et al., 2024). These results suggest that skill neologisms may provide a scalable path towards skill-based continual learning.

## 1. Introduction

Recent works have established that pretrained LLMs develop mastery over various skills and the ability to combine them beyond the pretraining distribution (Arora & Goyal, 2023; Chen et al., 2023; Yu et al., 2024). As LLMs are used to tackle an ever-growing range of problems, the ability to continuously grow model capabilities to new skills in a scalable fashion is a promising research direction.

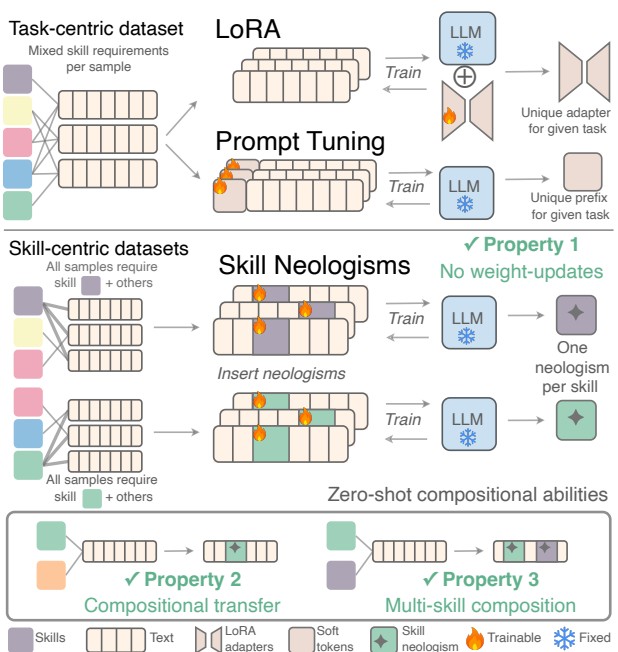

*Figure 1.* **Overview of Skill Neologisms.**

Yet, existing approaches to extend model capabilities fall short of this objective (Table 1). Finetuning models on new datasets risks catastrophic forgetting (Kirkpatrick et al., 2017; Luo et al., 2025), where previously mastered capabilities might disappear and safety risks might be introduced (Qi et al., 2024). In-context learning has shown some success at skill composition in simple settings (Chen et al., 2023; Levy et al., 2023; Xu et al., 2024), but it does not adapt as well as parameter-efficient fine-tuning (PEFT) methods (Liu et al., 2022a), and does not scale because of effective context limitations (Hsieh et al., 2024). Prompt tuning (Lester et al., 2021) can adapt models to new tasks by only learning soft tokens prepended to the prompt, and have been shown to rival full finetuning in some settings (Genewein et al., 2025). However, prefixes are typically task-specific instead of skill-specific, and learned prefixes cannot be composed or adapted to new settings without retraining (Asai et al., 2022; Wang et al., 2023).

[1]University of Cambridge. Correspondence to: Antonin Berthon <armb3@cam.ac.uk>.

*Proceedings of the 43rd International Conference on Machine Learning*, Seoul, South Korea. PMLR 306, 2026. Copyright 2026 by the author(s).

**Central question**

Can the compositional abilities of LLMs be leveraged to learn new composable skills without weight updates?

*Table 1.* Comparison of different approaches for skill-based continual learning. ‡E.g., full finetuning and LoRA (Hu et al., 2022). §E.g., prompt tuning (Lester et al., 2021) and related methods. *Achievable with skill-centered training.

| Method | P1: No Weight Updates | P2: Composes w/ OOD Skills | P3: Multi-Skill Composition |
|---|---|---|---|
| Finetuning-based‡ | ✗ | ✗ | ✗ |
| Prefix-based§ | ✓ | ✓* | ✗ |
| **Skill neologisms** | ✓ | ✓ | ✓ |

We term this objective *skill-based continual learning* and distinguish the following required properties: (▷ **P1**) New skills can be learned without modifying model parameters; (▷ **P2**) Learned skills are composable with other existing skills, including ones out-of-distribution from the training set; (▷ **P3**) Multiple skills learned independently can be composed without joint training.

In this work, we investigate whether **skill neologisms** might enable these properties (Figure 1). Inspired by neologisms proposed by Hewitt et al. (2025) for human-machine communication, skill neologisms aim to learn new vocabulary elements that, when provided in the model's context, improve the model capabilities on a specific skill. While sharing prompt tuning's core mechanism of optimizing soft tokens on a frozen model (P1), skill neologisms differ in two ways that are critical for composability:

- **Skill-centered training**. Training uses datasets where every sample requires the target skill, mixed with diverse skills already mastered by the model. Such datasets can be constructed in many settings, for example by leveraging the metacognitive capabilities of modern LLMs (Didolkar et al., 2024) (see Section 3.3).
- **Vocabulary-level integration**. Individual skills are learned via soft tokens (*skill tokens*) integrated in the model vocabulary, allowing the model to interact with the skill's procedural knowledge via in-context learning.

These two components encourage learning generally composable skill representations (P2), and enable zero-shot composition of independently learned skills (P3).

Our main contributions are as follows:

- We propose skill neologisms as a path toward skill-based continual learning (§ 3), and motivate this approach via empirical evidence that pretrained LLMs naturally exhibit vocabulary elements that encapsulate procedural knowledge (§ 4).
- We demonstrate in controlled settings (§ 5.2) that skill neologisms compose with OOD skills unseen during training

(P2) and enable zero-shot composition of independently learned skills (P3).

- We provide ablation experiments (§ 5.3) analyzing how token capacity and composition complexity in the training set affect learning of composable skill representations.
- We validate zero-shot composition of independently learned skill neologisms on the Skill-Mix benchmark (Yu et al., 2024), demonstrating their wider applicability to more realistic natural language settings (§ 6).

Code to reproduce our experiments can be found at `https://github.com/antoninbrthn/skill-neologisms`.

## 2. Preliminaries

### 2.1. Skills and Composition in Large Language Models

We build on the formalism introduced in Arora & Goyal (2023) in which skills refer to procedural knowledge—reusable capabilities such as arithmetic operations or logical reasoning—rather than factual knowledge. In this framework, any piece of text t is related to a set of skills S, and the understanding of text t requires mastery over all its underlying skills as well as their composition.

Given a set of skills $\Sigma$, we denote by $\mathcal{C}_k(\Sigma)$ the set of text pieces that require a $k$-tuple of skills from $\Sigma$. By extension, $\mathcal{C}_k(S_1, ., S_i, \Sigma)$ denotes text pieces that require *at least* skills $S_1, ., S_i$, mixed with $k - i$ other skills from $\Sigma$.

**Closed-form assumption** What does it mean to understand a text snippet $t$? A key assumption from Arora & Goyal (2023) is that the understanding of any piece of text can be tested via closed-form questions. This might be trivial if $t$ relates to a closed-form question (eg *"find the following number: 1,2,3,5,8,.."*), or by generating a set of multiple-choice questions as described in Arora & Goyal (2023).

**Composition beyond training** Modern LLMs demonstrate the ability to understand combinations of skills beyond their training distribution (Wei et al., 2022; He et al., 2024; Yu et al., 2024; Zhao et al., 2024). Theoretical analysis presented in Arora & Goyal (2023) links the emergence of skill composition ability to model scaling. Namely, scaling up model parameters by an order of magnitude leads to the same level of competence on $2k$-tuples of skills as the competence on $k$-tuples of the original model.

### 2.2. Soft Prompts and Prompt Tuning

**Soft tokens** Soft tokens $s = (s_1, ..., s_l)$ are sequences of continuous vectors of size $d_{\text{model}}$ (matching the model's hidden dimension), that can be inserted in a model's context after skipping the embedding matrix.

**Prompt tuning** Prompt-tuning (Lester et al., 2021) is a parameter-efficient finetuning approach where trainable soft

tokens $s$ are introduced as a prefix to the model's continuous representation of the input context. Only the soft tokens are optimized on a training set by back-propagating through the model while keeping the model's parameters frozen.

**Expressivity of prompt tuning** Recent works (Petrov et al., 2024; Genewein et al., 2025) study the conditions under which methods like prompt tuning might or might not succeed at learning a new task. Informally, a necessary condition is that the new task is *not too different* from tasks within the model's pretraining distribution, so that the model weights contain the necessary circuits to solve the new task.

### 2.3. Vocabulary Extensions via Neologisms

Neologisms embedding learning (Hewitt et al., 2025) uses learnable soft tokens as new vocabulary elements in a model's tokenizer and embedding matrix, that can then be used in prompts alongside text tokens. We denote a neologism of length $l$ as soft tokens $s = (s_1, ..., s_l)$, which extend the model's embedding matrix to $E' = E \cup s \in \mathbb{R}^{d_{\text{model}} \times (|\mathcal{V}|+l)}$ with columns $(s_1, ..., s_l)$, and its vocabulary to $l$ tokens: $\mathcal{V}' = \mathcal{V} \cup \{\langle S_1 \rangle, ..., \langle S_l \rangle\}$.

Like prompt tuning, the soft tokens of a neologism can be trained on samples that include the neologism tokens by back-propagating gradients through the frozen model. This is done via preference-based learning in Hewitt et al. (2025) but can be done similarly with supervised fine-tuning (SFT) or Reinforcement Learning Finetuning (RLFT).

## 3. Skill-based Continual Learning via Skill Neologisms

### 3.1. Skill-based Continual Learning: Problem Formulation

Given the composition capabilities of modern LLMs (Arora & Goyal, 2023; Yu et al., 2024; Zheng et al., 2024) as well as their in-context learning abilities (Wei et al., 2022), we investigate the following question: can LLMs learn new composable skills without weight updates? We term this objective *skill-based continual learning*, as it would allow models to acquire new composable skills without risk of catastrophic forgetting. For such an approach to be practical and scalable, it requires three key properties:

- **Property 1 (No weight updates)**: New skills are learned without modifying model parameters, preventing any catastrophic forgetting.

- **Property 2 (Compositional transfer)**: Learned skills compose with the model's existing skills in combinations not seen during training, including out-of-distribution skill combinations.

- **Property 3 (Multi-skill composition)**: Multiple indepen-

dently learned skills can be composed together zero-shot, without joint training on their combination.

Property 2 is necessary for the learned skill to be composable with skills held by the model beyond the training distribution, while Property 3 enables scalable continual learning where skills can be added incrementally and composed together even without joint training.

We propose *skill neologisms*—soft tokens integrated in the model's vocabulary—as one path towards achieving these properties. Our key hypothesis is that vocabulary-level interventions combined with skill-centered datasets can leverage the model's existing compositional abilities to learn composable representations of specific skills from the model's context.

### 3.2. Skill Neologisms: Overview

Skill neologisms are *soft tokens* integrated in the model vocabulary and optimized such that providing them in the model's context enhances the model's capability for a specific skill.

Figure 2 provides an overview of the different components required. We assume that a pretrained model $\mathcal{M}$ has mastered a set of skills $\Sigma$ and has the ability to compose them (Figure 2A). Our aim is to learn a new skill $S^*$. First, a skill-centered dataset $\mathcal{D}$ is constructed for skill $S^*$, with samples that all require at least skill $S^*$, as well as other skills from $\Sigma$ (Figure 2B). A skill neologism is initialized and added to the model's vocabulary and embeddings matrix. Then for each sample in $\mathcal{D}$, the neologism is inserted in the prompt and trained on $\mathcal{D}$ while keeping the rest of the model parameters fixed (Figure 2C).

### 3.3. Skill-centered datasets

Most datasets used for model pretraining or finetuning are *task-centered*: different samples or snippets of text implicitly depends on various skills. In contrast, skill neologisms require training on a dataset where every sample requires *at least* the target skill, mixed with other skills mastered by the model (Figure 1).

**Definition 3.1** (Skill-centered dataset). For a skill $S$ and set of skills $\Sigma$, an $S$-centered dataset is $\mathcal{D}(S, \Sigma) = \{t_i \sim \mathcal{C}_{k_i}(S, \Sigma)\}_i$, where each text snippet $t_i$ requires skill $S$ plus $k_i - 1$ additional skills sampled from $\Sigma$, with $k_i$ drawn from $\{1, \ldots, k_{\text{max}}\}$ according to some distribution $p$. We extend this notation to $(S_1, \ldots, S_m)$-centered datasets, where each snippet requires all of $S_1, \ldots, S_m$ plus up to $k_{\text{max}} - m$ additional skills from $\Sigma$.

**How to construct skill-centered datasets?** Since the skills that underlie samples are usually implicit, it may not be immediately obvious how to construct datasets centered

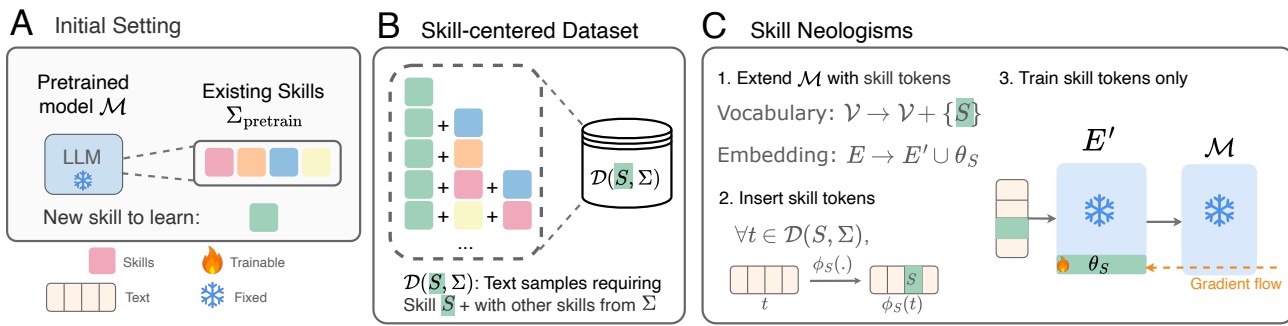

*Figure 2.* **Overview of Skill Neologisms.** (A) We consider pretrained model endowed with a set of implicit skills learned during pretraining. (B) A *skill-centered dataset* contains snippets of text that require *at least* the skill of interest, composed with pretraining skills. (C) *Skill neologisms* appends new token embeddings to the model's vocabulary and embedding matrix, which are trained on the skill-centered dataset while keeping the model parameters frozen. (D) By leveraging the pretrained model's compositional abilities, skill neologisms allow zero-shot composition with OOD skills, as well as composing independently learned skills.

around a specific skill. However, we note that this is possible in many practical settings. First, in structured or synthetic settings, the mapping between samples and skills is often explicit by construction. For example in the experiments presented Section 5, each sample (e.g. `[ASC][ADD]4165=2567`) maps naturally to the underlying skills (e.g. `[ASC]` and `[ADD]`). For more general settings, one can leverage the metacognitive abilities of strong LLMs to annotate samples with the implicit skills required (Didolkar et al., 2024), and then filter to examples that require at least the skill of interest. Finally, many datasets provide expertly curated multi-labels categorizing each data entry–such as in educational problem banks (Wang et al., 2020; Liu et al., 2023), programming benchmarks (Li et al., 2023), or reasoning tasks (Yuan et al., 2025)– which can be used to filter data around specific skills.

### 3.4. Skill Neologisms

**Definition 3.2** (Skill neologism)**.** Given a model $\mathcal{M}$ with parameters $\theta_{\text{LLM}}$, a *skill neologism* for skill $S$ is a set of learnable soft tokens (or *skill tokens*) $\theta_S \in \mathbb{R}^{d_{\text{model}} \times l}$ that minimizes some loss $\mathcal{L}$ over an $S$-centered dataset $\mathcal{D}(S, \Sigma)$:

$$\theta_S^* = \underset{\theta_S}{\text{argmin}} \, \mathbb{E}_{t \sim \mathcal{D}(S,\Sigma)}[\mathcal{L}(\mathcal{M}(\theta_{\text{LLM}}, \theta_S, \phi_S(t)))]$$

where $\phi_S : \text{Text} \to \text{Text}$ is an *insertion function* that inserts the skill neologism tokens into the text in a semantically appropriate way.

The loss function $\mathcal{L}$ depends on the training paradigm: cross-entropy loss for supervised fine-tuning (SFT), RL-style objectives (e.g., policy gradients) for reinforcement fine-tuning (RFT), or preference-based losses as in Hewitt et al. (2025). In our experiments, we use cross-entropy loss.

The choice of insertion function $\phi_S$ depends on the nature of the skill and how it naturally appears in text. We illustrate this with several examples below.

**Example of insertion functions** Depending on the under-

lying skill and text snippet, the insertion function might be simply replacing a given word with the neologism $s$ –as done in (Hewitt et al., 2025) by replacing *Ensure* by $Ensure_w^h$–, or introducing a short text such as "Make sure to use $\langle S \rangle$" (see Table 2 for examples under different settings).

*Table 2.* Examples of neologism insertion functions $\phi_s$. In each case the tokens corresponding to the skill neologism are shown in $\langle . \rangle$ brackets.

| Setting | Original text $t$ | Modified text $\phi_S(t)$ |
|---|---|---|
| Word replacement (Hewitt et al., 2025) | "Ensure that the length of the response is at least 600 words." | "$\langle Ensure_w^h \rangle$ that the length of the response is at least 600 words." |
| Word replacement (Section 5) | "[ADD][SHIFT]7283=..." | "[ADD]$\langle$SHIFT$\rangle$7283=..." |
| Task instruction | "Sort these numbers:" | "Sort these numbers using $\langle$SORT$\rangle$:" |

**Training procedure.** We outline the training procedure for skill neologisms in Algorithm 1.

---

**Algorithm 1** Training Skill Neologisms

**Require:** Pretrained model $\mathcal{M}$ with frozen parameters $\theta_{\text{LLM}}$
**Require:** Target skill $S$, set of pretrained skills $\Sigma$, skill-centered dataset $\mathcal{D}(S, \Sigma)$
**Require:** Skill neologism length $l$, insertion function $\phi_S$
**Require:** Learning rate $\eta$, number of epochs $T$
1: Initialize skill tokens $\theta_S \in \mathbb{R}^{d_{\text{model}} \times l}$
2: Extend vocabulary: $\mathcal{V}' \leftarrow \mathcal{V} \cup \{\langle S_1 \rangle, \dots, \langle S_l \rangle\}$
3: Extend embedding matrix: $E' \leftarrow E \cup \theta_S$
4: **for** $epoch = 1$ to $T$ **do**
5:    **for** each batch $\mathcal{B} \subset \mathcal{D}(S, \Sigma)$ **do**
6:       **for** each text sample $t \in \mathcal{B}$ **do**
7:          $t' \leftarrow \phi_S(t)$       # Insert skill tokens $\langle S_i \rangle$ into text
8:          Compute loss: $\mathcal{L} \leftarrow \text{CrossEntropy}(\mathcal{M}(\theta_{\text{LLM}}, \theta_S, t'))$
9:          Compute gradients: $\nabla_{\theta_S} \mathcal{L}$
10:       **end for**
11:       Update skill tokens: $\theta_S \leftarrow \theta_S - \eta \nabla_{\theta_S} \mathcal{L}$
12:       Keep model parameters $\theta_{\text{LLM}}$ unchanged
13:    **end for**
14: **end for**
15: **return** Optimized skill neologism $\theta_S^*$, extended vocabulary $\mathcal{V}'$

---

**Evaluating compositional transfer.** After training a skill neologism, we assess whether it has learned a general representation of skill $S$ rather than only fitting the compositions between $S$ and skills from $\Sigma_{\text{train}}$. We adopt the notion of competence from Arora & Goyal (2023) where a model's

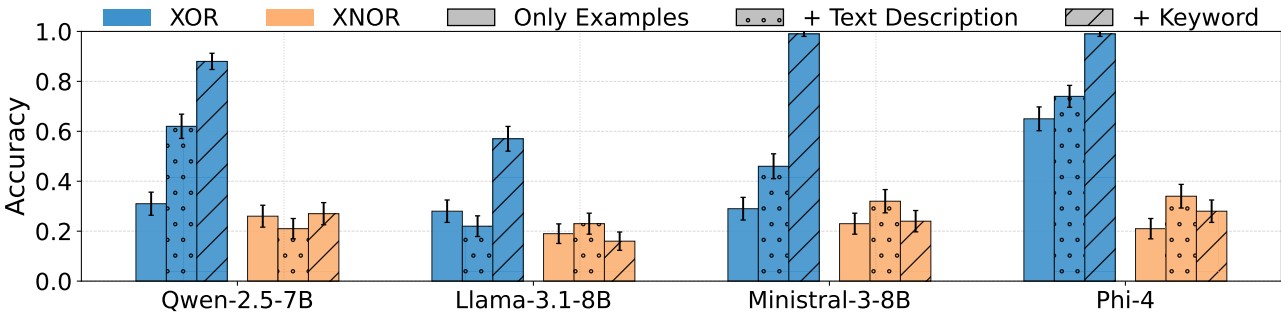

*Figure 3.* **Accuracy on XOR and XNOR completion tasks across open-source models under different prompts.** *Only Examples* provides three input–output examples before the query. + *Text Description* adds a natural-language description of the operation (e.g., "output 1 iff the input bits differ" for XOR). + *Keyword* adds only the operation name ("XOR" or "XNOR"). Results are averaged over $N = 100$ samples; error bars show standard error.

competence $\tau_S$ on a skill $S$ is its success rate on an $S$-centered dataset. We evaluate the neologism on two datasets: (i) in-distribution (ID) combinations involving skills $\Sigma_{\text{train}}$ denoted as $\tau_S^{\text{ID}}$; (ii) out-of-distribution (OOD) combinations involving held-out skills $\Sigma_{\text{test}}$, denoted as $\tau_S^{\text{OOD}}$.

A successful neologism should achieve $\tau_S^{\text{OOD}} \approx \tau_S^{\text{ID}}$, indicating that the skill neologisms composes with novel skills zero-shot.

### 3.5. Comparison to Existing Approaches

Common approaches to extend model capabilities, such as LoRA and prompt tuning-like methods, are typically trained on task-centric datasets. As a result, they learn task-specific patterns rather than generally composable skills, limiting out-of-distribution transferability (P2). Moreover, these approaches are structurally unable to achieve P3: independently trained adapters or prefixes cannot be combined without retraining on the target task.

Skill neologisms address this through two key components. First, skill-centered training with limited parameter capacity creates an inductive bias for learning generally composable skill representations (P2). Second, vocabulary-level integration leverages the model's in-context compositional abilities, allowing multiple independently learned skills to be combined simply by inserting multiple skill tokens in the context (P3).

## 4. Existence of Skill Tokens in Pretrained LLMs

Before training skill neologisms in Section 5, we first illustrate that pretrained LLMs already exhibit analogous behavior—some vocabulary tokens are associated with specific procedural knowledge. During pretraining, certain tokens are encountered in contexts related to particular operations. For example, "XOR" tokens will frequently appear in text discussing the corresponding logical operation, which is

analogous to a skill-centered dataset on the skill XOR. Consequently, these tokens might capture procedural knowledge for this operation. In contrast, less common tokens like "XNOR" might not—according to Google NGram Viewer, "XOR" appears approximately 15 times more frequently than "XNOR" in text from the past decade. We test this hypothesis on various open-source LLMs below.

**Setup** To test this hypothesis, we evaluate various open-source models on binary operation tasks. Models must perform XOR or XNOR on 3-bit sequences with 3 in-context examples. We compare accuracy across three conditions: (1) *Only examples*; (2) *Examples + description*: a textual description of the operation (e.g., "output 1 if and only if both input bits are different, and 0 otherwise" for XOR); and (3) *Examples + keyword*: only the keyword "XOR" or "XNOR". For conditions (2) and (3), information is inserted before examples via: "Complete the following using the skill:description/keyword". Each model is evaluated on $N = 100$ samples per setting.

**Results** Results are shown in Figure 3. For XOR, providing the keyword substantially improves accuracy over both other conditions, suggesting the "XOR" token has captured procedural knowledge through pretraining exposure, functioning as a genuine skill token. In contrast, for XNOR, neither keyword nor description improves accuracy beyond examples alone, indicating the "XNOR" token lacks sufficient training signal to encapsulate this operation. This demonstrates that skill tokens can emerge naturally when vocabulary tokens have sufficient exposure to skill-relevant contexts, and that they can encode procedural knowledge more efficiently than explicit descriptions.

> **Takeaway**
> Pretrained LLMs exhibit tokens that encode procedural knowledge, motivating the use of vocabulary-level parameters to learn skill representations.

# 5. Algorithmic Skill Composition

In this section, we evaluate skill neologisms on a controlled digit-sequence transformation task. We chose this setting because it provides explicit sample-skill definitions and un-ambiguous composition rules, unlike natural language tasks where skills are typically implicit. This enables us to construct exact ID/OOD splits over skills to cleanly measure whether the model learns general representations that compose with held-out skills (P2) and whether independently trained skills can be combined zero-shot (P3).

## 5.1. Setup

**Dataset** We create a synthetic dataset based on operations over digits sequences. Each sample is of the form: "[OP-1] ... [OP-k]x=y", where x is a random sequence of n digits, each OP-i is an operation and the output is the result of sequentially applying operations to x: $y = (\text{OP-k} \circ \dots \circ \text{OP-1})(x)$. Table 3 shows the different operations and example samples for $n = 3$.

*Table 3.* Digit-sequence transformation skills.

| Set | Skill | Description | Example (Seq. length: 3) |
|---|---|---|---|
| | ASC | Sort digits in ascending order | [ASC]472 = 247 |
| | DESC | Sort digits in descending order | [DESC]472 = 742 |
| | ADD | Add 1 to each digit | [ADD]472 = 583 |
| $\Sigma_{\text{pretrain}}$ | SUB | Subtract 1 from each digit | [SUB]472 = 361 |
| | REV | Reverse digit order | [REV]472 = 274 |
| | POL | Map odd (even) digits to 1 (0) | [POL]472 = 010 |
| | ID | Identity mapping | [ID]472 = 472 |
| $S_{\text{new}}$ | SHIFT | Right-shift digits | [SHIFT]472 = 247 |
| | INV-POL | Map odd (even) digits to 0 (1) | [INV-POL]472 = 101 |

**Base model** We fine-tune Qwen2.5-0.5B (Qwen et al., 2025) on $\mathcal{D}(\Sigma_{\text{pretrain}})$ with up to 3-skill combinations, using digit sequences of lengths $n \in [2, 9] \setminus \{5, 7, 9\}$, holding out lengths 5, 7, and 9 for validation. To ensure the model learns to combine operations flexibly, we also hold out 25% of 3-skill combinations. Training uses LoRA (Hu et al., 2022) in two phases: (i) 100k single-skill samples and (ii) 500k samples with $k = \{1, 2, 3\}$ drawn uniformly. Table 4 shows accuracy across skill counts and sequence lengths. The model achieves high accuracy on both in-distribution and held-out lengths, and generalizes well to unseen 3-skill combinations—with the exception of REV, which we exclude from $\Sigma_{\text{held-out}}$ due to overfitting on held-out lengths (see Appendix A1 and D.2.1).

**Learning new skills** We then freeze the pre-trained model $\mathcal{M}_{\text{pretrain}}$ and aim to learn two new skills $\Sigma_{\text{test}} = \{\text{SHIFT}, \text{INV-POL}\}$. For each skill $S_{\text{new}}$, we generate a dataset of 100k samples with 1-, 2-, and 3-combinations of $S_{\text{new}} \cup \Sigma_{\text{train}}$. To test out-of-distribution generalization, we create multiple leave-one-out datasets by setting $\Sigma_{train} = \Sigma_{\text{pretrain}} \setminus S_{\text{held-out}}$. This allows to test in-distribution on $\Sigma_{train}$ and out-of-distribution on $S_{\text{held-out}}$.

**Models** For each new skill $S_{\text{new}} \in \Sigma_{\text{test}}$ and held-out

*Table 4.* Accuracy of $\mathcal{M}_{pretrain}$ over sequence lengths for single-task ($\mathcal{C}_1$), two-task ($\mathcal{C}_2$), and three-task ($\mathcal{C}_3$) compositions. * indicates lengths and combinations held-out during training. ID: in-distribution skill combinations, OOD: out-of-distribution skill combinations. Per-operation details shown in Appendix A1.

| | Composition setting | | | |
|---|---|---|---|---|
| **Sequence length** | $\mathcal{C}_1(\Sigma_{\text{pretrain}})$ | $\mathcal{C}_2(\Sigma_{\text{pretrain}})$ | $\mathcal{C}_3(\Sigma_{\text{pretrain}})$ | |
| | **ID** | **ID** | **ID** | **OOD*** |
| 2 | 100.0% | 100.0% | 100.0% | 97.0% |
| 3 | 100.0% | 100.0% | 100.0% | 96.0% |
| 4 | 99.1% | 100.0% | 98.0% | 97.0% |
| 5* | 97.6% | 98.0% | 95.0% | 84.0% |
| 6 | 95.6% | 94.0% | 95.0% | 89.0% |
| 7* | 92.6% | 92.0% | 89.0% | 74.0% |
| 8 | 92.6% | 90.0% | 79.0% | 74.0% |
| 9* | 83.9% | 75.0% | 74.0% | 58.0% |

skill $S_{\text{held-out}} \in \Sigma_{\text{pretrain}}$, we train three model variants on 100k samples from $D(S_{\text{new}}, \Sigma_{\text{pretrain}} \setminus S_{\text{held-out}})$ with up to $k_{\max} = 3$ operations: (1) **Skill neologisms** with length $l = 20$, initialized from the mean embedding of $\Sigma_{\text{pretrain}}$ operation tokens; (2) **Prompt tuning** (Lester et al., 2021) with a prefix of length $l = 20$ using the same initialization; (3) **LoRA** (Hu et al., 2022) with rank $r = 16$.

## 5.2. Results

We now validate that skill neologisms satisfy P2 and P3 from Section 3.1. P1 (no weight updates) is satisfied by construction, as all model parameters are frozen when training skill neologisms. We evaluate P2 by testing whether learned skills compose with held-out skills not seen during training, and P3 by testing whether independently learned skill neologisms can be combined zero-shot.

### 5.2.1. PROPERTY 2: COMPOSITIONAL TRANSFER

Figure 4 shows the accuracy of LoRA, prompt tuning, and skill neologisms on 2-combinations of $S_{\text{new}}$ with either skills from $\Sigma_{\text{train}}$ (in-distribution) or $S_{\text{held-out}}$ (out-of-distribution). All three methods achieve near-perfect in-distribution accuracy. However, only skill neologisms consistently succeed at composing $S_{\text{new}}$ with $S_{\text{held-out}}$. LoRA shows the poorest OOD generalization, suggesting it overfits the training distribution rather than learning a composable representation of $S_{\text{new}}$. Prompt tuning performs intermediately; the gap with skill neologisms is notable given both optimize the same number of soft tokens. This suggests that semantically embedding the soft tokens inside the prompts may provide additional flexibility to learn composable skill representation. Accuracy on 3-combinations shows similar patterns (Figure A2 in Appendix).

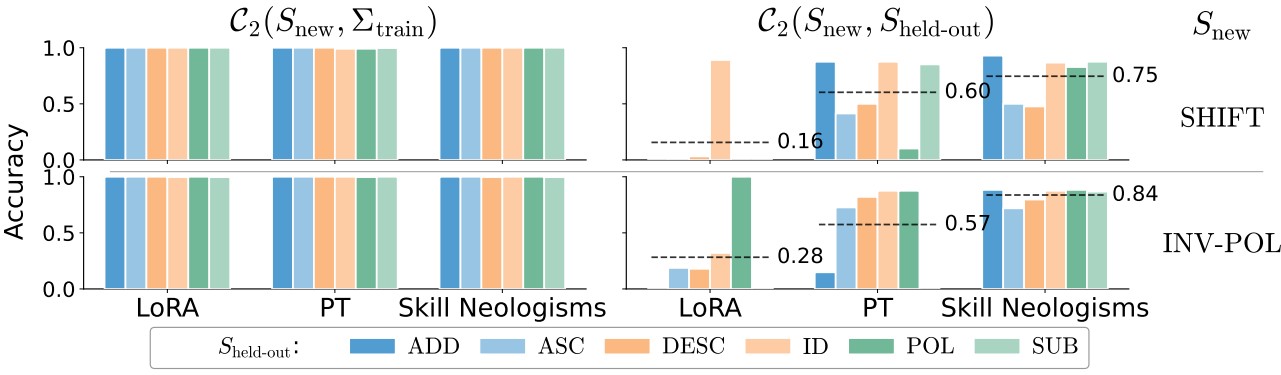

*Figure 4.* Accuracy on 2-combinations of skills mixing $S_{\text{new}}$ with $\Sigma_{\text{train}}$ (in-distribution) or $S_{\text{held-out}}$ (out-of-distribution). Dotted lines show the average accuracy across all $S_{\text{held-out}}$. PT: Prompt Tuning.

> **Takeaway**
> Skill neologisms learn composable skill representations that transfer to OOD compositions.

### 5.2.2. PROPERTY 3: MULTI-SKILL COMPOSITION

We test whether the skill neologisms learned independently for SHIFT and INV-POL in § 5.2.1 can be combined zero-shot to handle compositions requiring both skills (Property 3). This distinguishes skill neologisms from LoRA and prompt tuning-like approaches, which cannot be composed after independent training without retraining on the joint task. We compare against in-context learning (ICL)–a natural baseline for zero-shot composition–by providing $\mathcal{M}_{\text{pretrain}}$ with $N \in \{10, 20, 50, 100\}$ examples from $\mathcal{D}(S_{\text{new}}, \Sigma_{\text{pretrain}})$ for $S_{\text{new}} \in \{\text{SHIFT}, \text{INV-POL}\}$ ($2N$ examples in total).

Figure 5 shows the average accuracy across different $S_{\text{held-out}}$ for skill neologisms and $N$ for ICL, for different sequence lengths (increasing task difficulty). Traces for individual runs are shown Figure A3 in the Appendix. Skill neologisms significantly outperform ICL across all sequence lengths. This demonstrates that skill neologisms successfully capture reusable procedural knowledge that transfers zero-shot to new compositions, whereas ICL struggles to extract and combine the relevant patterns from examples alone.

> **Takeaway**
> Independently learned skill neologisms can be successfully composed zero-shot.

### 5.3. Insights and Ablation Experiments

Having validated that skill neologisms satisfy Properties 2 and 3, we study and ablate different components to understand the mechanisms at play. We focus on three questions: (1) How does the capacity of skill tokens affect their abil-

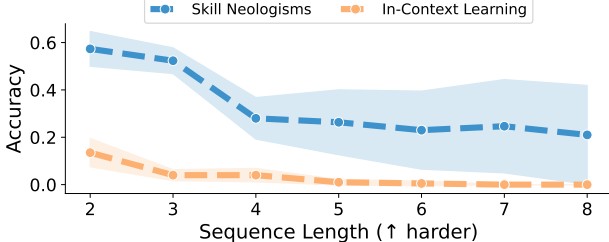

*Figure 5.* **Zero-shot composition of SHIFT and INV-POL.** *Skill Neologisms*: we compose the skill tokens learned independently for SHIFT and INV-POL for a given $S_{\text{held-out}}$, and plot the average accuracy (±std) across the 6 $S_{\text{held-out}}$. *In-Context Learning*: we provide in-context $N = \{10, 20, 50, 100\}$ examples sampled from $\mathcal{D}(S_{\text{new}}, \Sigma_{\text{pretrain}})$ for $S_{\text{new}} \in \{\text{SHIFT}, \text{INV-POL}\}$ ($2N$ examples in total), and plot the average accuracy(±std) across the 4 runs.

ity to learn composable representations? (2) How does the diversity of skill combinations in training data impact generalization? (3) Is performance sensitive to initialization and label noise?

### 5.3.1. SKILL NEOLOGISM LENGTH

**Motivation** We hypothesize that limited capacity of skill tokens provides an inductive bias to learn generally composable representations rather than overfitting to the training distribution, as we observed with LoRA in Figure 4. This suggests a trade-off on the length of the skill neologism: too few parameters may fail to learn the skill, while too many may reduce generalization to OOD compositions.

**Setup** We train skill neologisms for SHIFT and INV-POL with $S_{\text{held-out}} = \text{ADD}$, varying the neologism length from $l = 1$ ($|\theta_S| = 768$ parameters) to $l = 200$ ($|\theta_S| = 153k$ parameters).

**Results** Figure 6 shows the accuracy on 2-combinations with $\Sigma_{\text{train}}$ (ID) and $S_{\text{held-out}}$ (OOD) for varying skill lengths. The model gets near-perfect accuracy in-

distribution for $l \geq 5$. However, the accuracy out-of-distribution first increases with higher capacity, but then drops as $l$ becomes too large ($l > 20$). This suggest that after a certain point, increased capacity for the skill tokens becomes detrimental to learning a generally composable representation of the skill.

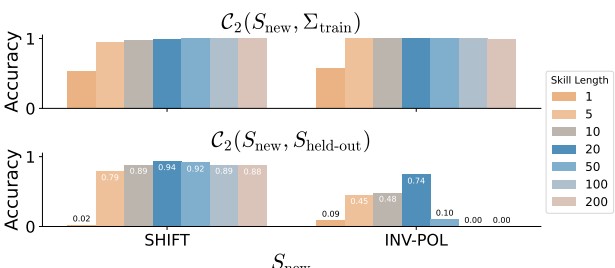

*Figure 6.* **Effect of skill token length.** Accuracy on 2-skill combinations with $S_{\text{held-out}} = \texttt{ADD}$ for varying skill token length $l$.

> **Takeaway**
> The limited capacity of skill neologisms acts as an inductive bias to learn more composable skill representations.

### 5.3.2. COMPOSITION COMPLEXITY IN TRAINING SET

The complexity of skill combinations in the training set provides another source of inductive bias: exposing skill tokens to the target skill in more complex compositions (higher $k_{\max}$) may improve their ability to compose with held-out skills.

**Setup** We train neologisms for $\texttt{SHIFT}$ and $\texttt{INV-POL}$ for various $S_{\text{held-out}}$, while varying the maximum number of compositions $k_{\max} \in \{1, 2, 3\}$ in the training set, keeping the total number of samples fixed at 100k. We compare the OOD accuracy on 2- and 3-compositions involving the held-out skill.

**Results** Table 5 shows the accuracy averaged across all $S_{\text{held-out}}$ (see Appendix A.4 for detailed results across $S_{\text{held-out}}$). Training on more compositions in the training data generally improves OOD generalization. In particular, 2-skill compositions benefit from having been trained on 3-skill composition data for $\texttt{INV-POL}$.

### 5.3.3. ROBUSTNESS TO INITIALIZATION AND LABEL NOISE

**Effect of initialization** Prompt tuning-like methods are known to depend on initialization (Lester et al., 2021). Table 6 compares random initialization against initialization from the average embedding of tokens in $\Sigma_{\text{pretrain}}$ (see Appendix A.5 for detailed results across $S_{\text{held-out}}$). Initialization from pretrained skill embeddings shows marginally better performance (particularly for $\texttt{INV-POL}$ on 2-skill

*Table 5.* **Effect of composition complexity in the training set.** OOD accuracy(±std) on 2-skill and 3-skill combinations when training with varying maximum composition complexity $k_{\max}$. Results averaged across all held-out skills $S_{\text{held-out}}$ (see Appendix A.4 for a detailed breakdown across $S_{\text{held-out}}$).

| $S_{new}$ | $k_{\max}$ | $\mathcal{C}_2(S_{new}, S_{\text{held-out}})$ | $\mathcal{C}_3(S_{new}, S_{\text{held-out}}, \Sigma_{\text{train}})$ |
|---|---|---|---|
| INV-POL | 1 | $.36 \pm .44$ | $.05 \pm .06$ |
| | 2 | $.60 \pm .34$ | $.61 \pm .24$ |
| | 3 | $.85 \pm .06$ | $.65 \pm .23$ |
| SHIFT | 1 | $.46 \pm .04$ | $.55 \pm .04$ |
| | 2 | $.72 \pm .21$ | $.61 \pm .19$ |
| | 3 | $.72 \pm .18$ | $.69 \pm .14$ |

compositions), but skill tokens trained from random initialization still show strong OOD composition abilities.

*Table 6.* **Effect of initialization.** OOD accuracy(±std) on 2-skill and 3-skill combinations for random initialization versus initialization from average embeddings of pretrained skills $\Sigma_{\text{pretrain}}$. Results averaged across all held-out skills $S_{\text{held-out}}$.

| $S_{new}$ | Init Method | $\mathcal{C}_2(S_{new}, S_{\text{held-out}})$ | $\mathcal{C}_3(S_{new}, S_{\text{held-out}}, \Sigma_{\text{train}})$ |
|---|---|---|---|
| INV-POL | From $\Sigma_{\text{pretrain}}$ | $.85 \pm .06$ | $.65 \pm .23$ |
| INV-POL | Random | $.63 \pm .32$ | $.58 \pm .31$ |
| SHIFT | From $\Sigma_{\text{pretrain}}$ | $.72 \pm .18$ | $.69 \pm .14$ |
| SHIFT | Random | $.70 \pm .19$ | $.65 \pm .14$ |

**Effect of label noise** Accurately curating skill-centered datasets might not always be possible, resulting in noisy skill labels. To evaluate the impact of noise on skill labels, we train skill neologisms for $\texttt{INV-POL}$ with $S_{\text{held-out}} = \texttt{ADD}$ while randomly replacing occurrences of the target skill in the training set with a random other skill and updating the ground-truth answer accordingly. This results in the skill neologism being trained on a dataset that sometimes involves an operation that is not the expected skill. The results are presented Figure A4 in the Appendix, and show that while the 1-op accuracy is robust up to around $40\%$ of erroneous labels, noise in the skill labels degrades 2-op and 3-op performance with $20\%$ noise, particularly for OOD compositions.

## 6. Natural Language Skill Composition

To demonstrate the applicability of skill neologisms to a realistic language setting, we conduct experiments on the Skill-Mix benchmark (Yu et al., 2024), which evaluates the ability of LLMs to combine natural language skills. For each sample, the model is asked to generate a short text that illustrates a list of language skills (e.g. "metaphor" or "syllogism") on a specific topic. Then, the produced text is graded by a strong LLM (here, GPT-5) based on whether or not it includes the desired skills. While this benchmark was introduced with the aim to measure LLMs' ability to combine skills, it provides a useful testbed to evaluate a model's mastery and acquisition of different skills. In our

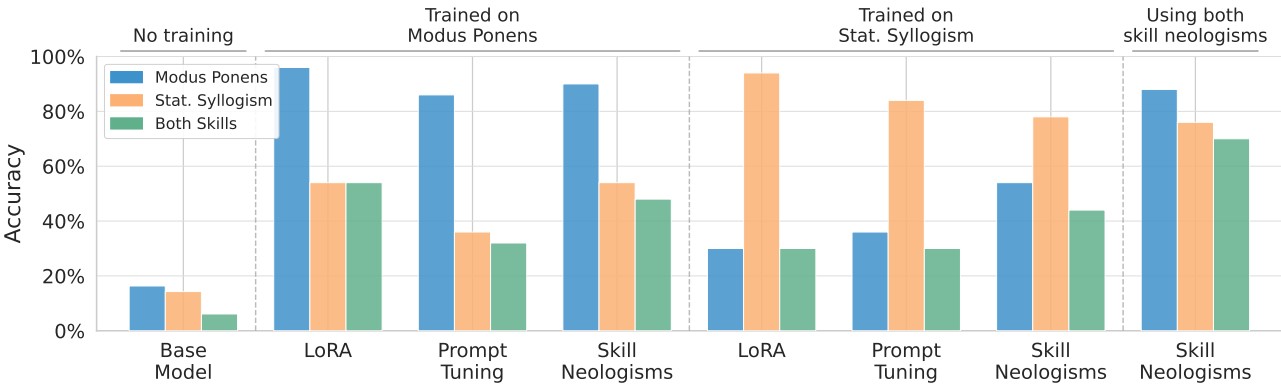

*Figure 7.* **Learning two target skills on Skill-Mix**. "No training": vanilla Llama3.2-3B-Instruct model. "Trained on $S_{new}$": each model is trained on a $S_{new}$-centered dataset mixing $S_{new}$ with other skills. "Using both skill neologisms": combining the independently learned skill tokens zero-shot. "Both Skills" corresponds to successfully producing a text that includes both skills. Grading is done with GPT-5. Prompts for inference and grading follow the official Skill-Mix repository.

case, we use this setting to demonstrate that neologisms learned independently for different skills can be successfully composed zero-shot.

**Setup** We run Llama3.2-3B-Instruct on the Skill-Mix repository (containing 10 publicly released skills) and select two skills for which the base model has low accuracy: "modus ponens" and "statistical syllogism". For each target skill, we construct N=300 training samples by mixing the target skill with 0, 1 or 2 randomly sampled skills (while holding out the other target skill) and collecting high-quality answers using GPT-5. We train with LoRA, prompt tuning and skill neologisms via SFT and test on 50 queries requiring both target skills simultaneously. We use the prompts from the Skill-Mix repository, which include a definition and one example for each queried skill (detailed hyperparameters and prompts are provided in Appendix D.3). For skill neologisms, we use $l = 20$ and insert neologisms into the prompt by replacing every occurrence of the skill name with the corresponding soft tokens.

**Results** Figure 7 shows the accuracy of each method after being trained on either one of the target skills. LoRA, prompt tuning and skill neologisms reach similarly high accuracy for the skill they were trained for, but not for the other held-out target skill. Skill neologisms allow for the zero-shot combination of both independently learned skill tokens, achieving high accuracy on both skills simultaneously despite never being trained on them jointly.

## 7. Discussion

**Related Work** Our work relates to three main research directions (detailed comparison in Appendix B). First, prior research has investigated skills and compositional abilities in LLMs using in-context skill descriptions (Chen et al., 2023), skill-rich synthetic data (Zhao et al., 2024; Kaur et al., 2025), or skill-targeted training (He et al., 2026). In contrast, we

learn generally composable skill representations via soft tokens integrated into the model vocabulary. Second, while prefix-based adaptation methods (Lester et al., 2021; Li & Liang, 2021) and their transferability extensions (Vu et al., 2022; Wang et al., 2023) improve generalization across tasks, they ultimately require training on the target task. We instead adopt a skill-centric perspective, focusing on out-of-distribution generalization and zero-shot composition of independently learned skills. Finally, meaningful soft tokens have been studied for visual concepts (Gal et al., 2023), tool representations (Hao et al., 2023), prompt compression (Mu et al., 2023; Sastre & Rosá, 2025; Kuratov et al., 2025), and behavioral alignment (Radevski et al., 2026). To the best of our knowledge, our work is the first to propose learning composable soft tokens that encapsulate specific procedural knowledge.

**Limitations and Future Work** Our work constitutes an initial proof-of-concept of skill neologisms as a path towards skill-based continual learning, focusing on a controlled experimental setting and a natural language task where skill can be well-defined. Further work is needed to explore skill neologisms in other settings where the definition of skills and their compositions is fuzzier. Key challenges include the construction and availability of diverse skill-centered datasets, as well as the optimization instability inherent to training soft tokens (see detailed discussion in Appendix C).

**Conclusion** We propose skill neologisms as a way to extend LLM capabilities to specific skills by optimizing vocabulary-integrated soft tokens on skill-centered data. By demonstrating compositional transfer to out-of-distribution skills and zero-shot combination of independently learned soft tokens, both in a controlled algorithmic task and a more realistic natural language setting, our findings confirm that skill neologisms are a promising direction for scalable skill-based continual learning.

## Impact Statement

This paper presents work whose goal is to advance the field of machine learning. Skill neologisms are a method for extending LLM capabilities without weight updates, and their societal implications are broadly similar to those of LLMs and parameter-efficient fine-tuning methods more generally. We do not identify specific consequences that we feel need to be highlighted here.

## Acknowledgments

This work was supported by Azure sponsorship credits granted by Microsoft's AI for Good Research Lab. AB's research is supported by funding from Eedi, and NA is sponsored by W.D. Armstrong Trust.

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

# A. Extended Results

### A.1. Model pre-training

Figure A1 shows the accuracy of $\mathcal{M}_{\text{pretrain}}$ after pre-training (same as Table 4), across sequence lengths and operations. Sequence lengths $\{2, 3, 4, 6, 8\}$ are in-distribution, while lengths $\{5, 7, 9\}$ were held-out from pre-training data. The model successfully learns most operations over the training distribution and generalizes to unseen sequence lengths. The only exception is REV, which does not generalize to OOD sequence lengths.

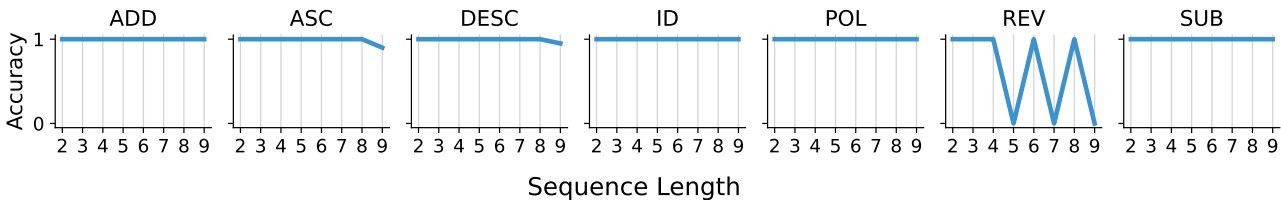

*Figure A1.* Accuracy of $\mathcal{M}_{\text{pretrain}}$ sequence lengths for each pre-train operations. Sequence lengths $\{2, 3, 4, 6, 8\}$ are in-distribution, while lengths $\{5, 7, 9\}$ were held-out from pre-training data.

### A.2. Out-of-distribution generalization

Following the experimental setup from Section 5.2.1, Figure A2 shows the ID and OOD accuracy on 3-compositions of skills. For OOD, samples are drawn from $\mathcal{C}_3(S_{new}, S_{\text{held-out}}, \Sigma_{\text{train}})$, where $S_{\text{new}}$ and $S_{\text{held-out}}$ are always included and one operation from $\Sigma_{\text{train}}$ is sampled, and the order of the three operations is randomly permuted.

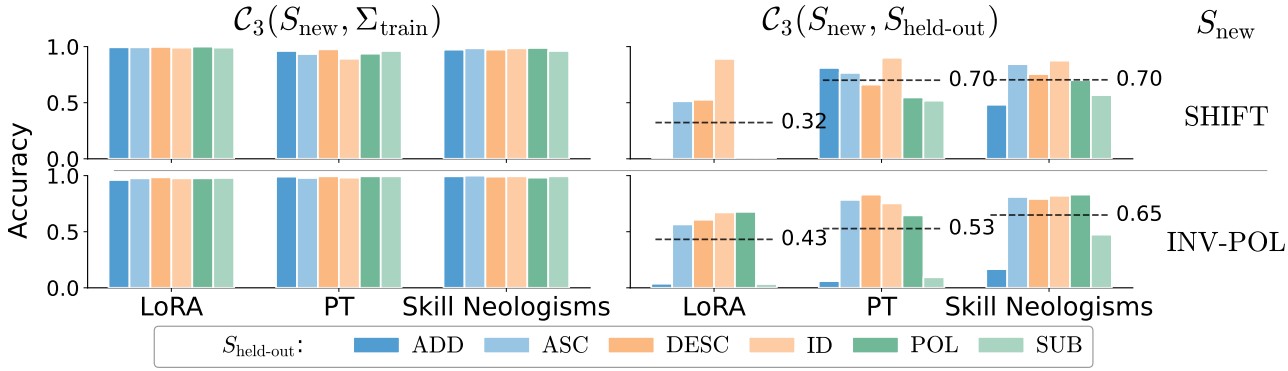

*Figure A2.* Accuracy on 3-combinations of skills mixing $S_{\text{new}}$ with $\Sigma_{\text{train}}$ (in-distribution) or $S_{\text{held-out}}$ (out-of-distribution). Dotted lines show the average accuracy across all $S_{\text{held-out}}$. PT: Prompt Tuning.

### A.3. Multi-skill composition

Figure A3 shows detailed results from Section 5.2.2 with accuracy on individual pairs of neologisms (for a given $S_{\text{held-out}}$) for skill neologisms, and individual number of examples $N$ for ICL.

### A.4. Effect of compositions in training set

Table A1 shows the detailed accuracy across $S_{\text{held-out}}$ skills for the experiment presented Section 5.3.2.

### A.5. Effect of initialization

Table A2 shows the detailed accuracy across $S_{\text{held-out}}$ skills for the initialization ablation presented Section 5.3.3.

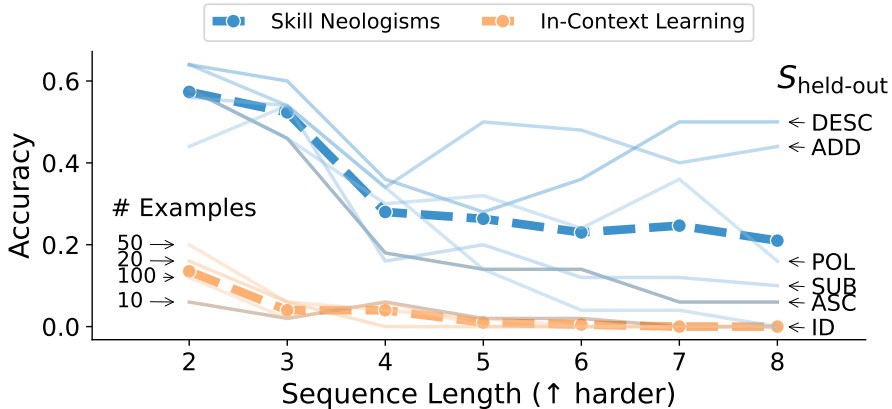

*Figure A3.* **Zero-shot composition of `SHIFT` and `INV-POL`.** *Skill Neologisms*: we compose the skill tokens learned independently for `SHIFT` and `INV-POL` for a given $S_{\text{held-out}}$ (thin blue lines), and plot the average accuracy (±std) across the 6 $S_{\text{held-out}}$ (thick dashed blue line). *In-Context Learning*: we provide in-context $N = \{10, 20, 50, 100\}$ examples sampled from $\mathcal{D}(S_{\text{new}}, \Sigma_{\text{pretrain}})$ for $S_{\text{new}} \in \{\text{SHIFT}, \text{INV-POL}\}$ ($2N$ examples in total), and plot individual results (thin orange lines) and the average (thick dashed orange line) across the 4 runs. Thin lines show the individual runs across $S_{\text{held-out}}$ and $N$.

*Table A1.* **Effect of number of compositions in training set.**

| | | \multicolumn{7}{c}{Acc on $\mathcal{C}_2(S_{new}, S_{\text{held-out}})$} | \multicolumn{7}{c}{Acc on $\mathcal{C}_3(S_{new}, S_{\text{held-out}}, \Sigma_{\text{pretrain}})$} |
| | | ASC | DESC | ADD | SUB | ID | POL | **AVG** | ASC | DESC | ADD | SUB | ID | POL | **AVG** |
| $S_{new}$ | k-ops | | | | | | | | | | | | | | |
|---|---|---|---|---|---|---|---|---|---|---|---|---|---|---|---|
| INV-POL | 1 | .15 | .06 | .00 | .00 | .99 | .95 | .36 | .07 | .03 | .00 | .00 | .02 | .18 | .05 |
| | 2 | .43 | .80 | .46 | .00 | .90 | .98 | .60 | .76 | .81 | .33 | .22 | .83 | .73 | .61 |
| | 3 | .73 | .80 | .88 | .86 | .89 | .91 | .85 | .83 | .78 | .21 | .47 | .81 | .82 | .65 |
| SHIFT | 1 | .48 | .51 | .40 | .42 | .46 | .51 | .46 | .58 | .62 | .51 | .53 | .53 | .56 | .55 |
| | 2 | .40 | .50 | .95 | .92 | .84 | .71 | .72 | .69 | .76 | .38 | .37 | .90 | .58 | .61 |
| | 3 | .52 | .46 | .92 | .87 | .87 | .70 | .72 | .82 | .73 | .49 | .54 | .88 | .70 | .69 |

*Table A2.* **Effect of initialization.** OOD accuracy on 2-skill and 3-skill combinations for random initialization versus initialization from average embeddings of pretrained skills $\Sigma_{\text{pretrain}}$.

| | | \multicolumn{7}{c}{Acc on $\mathcal{C}_2(S_{new}, S_{\text{held-out}})$} | \multicolumn{7}{c}{Acc on $\mathcal{C}_3(S_{new}, S_{\text{held-out}}, \Sigma_{\text{pretrain}})$} |
| | | ADD | ASC | DESC | ID | POL | SUB | **AVG** | ADD | ASC | DESC | ID | POL | SUB | **AVG** |
| $S_{new}$ | Init Method | | | | | | | | | | | | | | |
|---|---|---|---|---|---|---|---|---|---|---|---|---|---|---|---|
| INV-POL | From $\mathcal{S}_{pretrain}$ | .88 | .73 | .80 | .89 | .91 | .86 | .85 | .21 | .83 | .78 | .81 | .82 | .47 | .65 |
| | Random | .44 | .80 | .81 | .88 | .88 | .00 | .63 | .23 | .80 | .79 | .78 | .83 | .05 | .58 |
| SHIFT | From $\mathcal{S}_{pretrain}$ | .92 | .52 | .46 | .87 | .70 | .87 | .72 | .49 | .82 | .73 | .88 | .70 | .54 | .69 |
| | Random | .88 | .46 | .41 | .87 | .70 | .85 | .70 | .50 | .77 | .71 | .87 | .54 | .50 | .65 |

## A.6. Effect of noisy skill labels

Figure A4 shows the impact of varying magnitudes of noise on skill labels on downstream accuracy, using $l = 5$, $S_{new} = \text{INV-POL}$ and $S_{\text{held-out}} = \text{ADD}$.

## B. Extended Related Work

**Skills and compositional abilities in LLMs** Recent works have proposed ways to extend model capabilities to new skills and compositions. Skill-in-Context (Chen et al., 2023) aims to elicit compositional abilities in LLMs by providing in-context descriptions of skills and step-by-step explanations on how to compose them. Zhao et al. (2024) show that training LLMs on skill-rich synthetic datasets improve compositional abilities, even on held-out skills unseen during training. STAT (He et al., 2026) aims to improve model capabilities by uncovering specific skills lacking from the model, and targeting these skills via either reweighting or synthetic data augmentations. Didolkar et al. (2024) demonstrated that LLMs have the ability

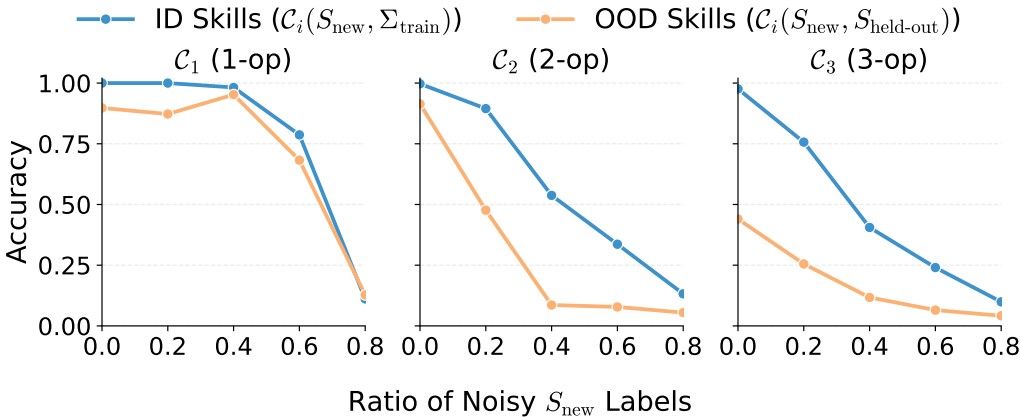

*Figure A4.* **Effect of noise in the skill-centered train set.** Underlying operations labeled as the target skills $S_{new}$ are corrupted to another operation with varying proportions in the training set. Using parameters $l = 5$, $S_{new} = \text{INV-POL}$ and $S_{held-out} = \text{ADD}$. Increasing label noise on the target skill gradually degrades performance, particularly for 2- and 3-skill compositions and OOD skills.

to describe skills required by a given task, while Kaur et al. (2025) leveraged such metacognition abilities of LLMs to create a skill-rich synthetic dataset for instruction-tuning. In contrast, we propose learning generally composable representation of skills via soft tokens, allowing composition with other skills thanks to the pre-trained model's general compositional abilities.

**Prefix-Based Adaptations** Prompt tuning (Lester et al., 2021) first introduced the paradigm of training soft tokens appended to the input prompt to adapt a frozen model to new tasks, which P-Tuning (Liu et al., 2024) extended by mixing soft prompts produced by a prompt encoder with discrete text tokens. Concurrently, prefix tuning (Li & Liang, 2021) proposed learning prefix key and value vectors at every layer of the model–yielding more expressive power than the input layer only–which P-Tuning v2 (Liu et al., 2022b) extended to natural language understanding (NLU) settings. Several works have focused on enhancing the transferability of prompt tuning. SPoT (Vu et al., 2022) train prompts across diverse tasks to improve transferability to new ones; Multitask Prompt Tuning (MTP) (Wang et al., 2023) decomposes prompts between shared and task-specific components; ATTEMPT (Asai et al., 2022) combines prompts from different tasks using an attention mechanism. However, these methods still require training on the target task, unlike skill neologisms that can combine independently learned soft-prompts zero-shot.

**Meaningful Soft Tokens** Another line of research has focused on learning soft tokens with specific, grounded meanings, moving beyond their use as purely task-specific adapters. In the vision-language domain, Textual Inversion (Gal et al., 2023) learns a new pseudo-word in the embedding space of a frozen model to represent a novel visual concept, such as a specific object or artistic style. In function calling and tool use for LLMs, ToolkenGPT (Hao et al., 2023) represents tools via tokens integrated in the model vocabulary. In prompt compression, memory tokens (Sastre & Rosá, 2025; Kuratov et al., 2025) compress long sequences of text into a single reversible embedding, while gist tokens (Mu et al., 2023) replace prompts with gist tokens that preserve downstream model behavior. Recently, Radevski et al. (2026) proposed learning composable steering tokens for behavioral alignment. To the best of our knowledge, our work is the first to learn composable soft tokens that encapsulate specific procedural knowledge.

## C. Extended Limitations

**Skill-centered dataset construction** While we argue that skill-centered datasets can be identified or constructed in a variety of contexts, their availability for a given skill remains a key requirement for learning skill neologisms. Moreover, as our experiments suggest, the quality of the learned neologisms partly depends on the complexity of the data and on how the target skill is mixed with a diverse set of other skills during training. Assessing and ensuring such diversity may not be straightforward in all settings.

**Soft token limitations** Skill neologisms rely on optimizing soft tokens, in a manner similar to prompt tuning. As a result, they inherit several limitations commonly associated with prompt tuning, including sensitivity to initialization and to

hyperparameters such as token length and learning rate. In addition, successfully learning soft tokens requires that the target task remains reasonably close to the model's pretraining distribution, as shown in Petrov et al. (2024) and Genewein et al. (2025).

**Computational cost** Although soft tokens are substantially more parameter-efficient than standard fine-tuning methods, training them still requires backpropagation through the full model. This leads to computational costs that can be comparable to those of fine-tuning in practice. As a result, training skill neologisms for large-scale models (e.g., > 30B parameters) may remain challenging without access to substantial computational resources.

# D. Experimental Details

Code to reproduce our experiments can be found at `https://github.com/antoninbrthn/skill-neologisms` and `https://github.com/vanderschaarlab/skill-neologisms`.

## D.1. Experimental Details: Section 4

### D.1.1. EXPERIMENTAL SETUP

**Models Evaluated:**

- Qwen2.5-7B (Qwen et al., 2025)

- Llama-3.1-8B (Grattafiori et al., 2024)

- Ministral-3-8B-Base-2512 (Liu et al., 2026)

- Phi-4 (Abdin et al., 2024)

**Tasks:** Binary operations XOR and XNOR on 3-bit sequences.

**Dataset Configuration:**

- Test samples: 100 per task (XOR, XNOR)

- Bit length: 3

- In-context examples: 3 examples per prompt

- Example format: Each sample contains 3 input-output pairs followed by a query input

**Prompt Variations:** Three prompt formulations were tested for each task:

1. **Only Examples (Baseline):** No additional context provided, only the 3 in-context example pairs

2. **Examples + Keyword:** A symbolic keyword prefix is added before the examples
   - XOR: "Complete the following using the skill: 'XOR' "
   - XNOR: "Complete the following using the skill: 'XNOR' "

3. **Examples + Text Description:** A natural language description is provided
   - XOR: "Complete the following using the skill: 'output 1 iif both input bits are different, and 0 otherwise' "
   - XNOR: "Complete the following using the skill: 'output 1 iif both input bits are the same, and 0 otherwise' "

**Example Prompt Structure:**

For the "Examples + Keyword" variant (XOR):

```
Complete the following using the skill: 'XOR'
101 011 = 110
100 110 = 010
011 001 = 010
111 010 =
```

For the "Only Examples" variant:

```
101 011 = 110
100 110 = 010
011 001 = 010
111 010 =
```

For the "Examples + Text Description" variant (XOR):

```
Complete the following using the skill: 'output 1 iif
both input bits are different, and 0 otherwise'
101 011 = 110
100 110 = 010
011 001 = 010
111 010 =
```

### D.1.2. EVALUATION DETAILS

**Inference Parameters:**

- Batch size: 16

- Generation method: Greedy decoding (deterministic)

- Padding side: left

- Models run in evaluation mode

**Metrics:**

- **Exact Match Accuracy:** Percentage of test samples where the model's generated output exactly matches the ground truth

- **Standard Error:** Computed assuming binomial distribution: $\text{SE} = \sqrt{\frac{p(1-p)}{N}}$ where $p$ is accuracy and $N = 100$

### D.2. Experimental Details: Section 5

This appendix provides comprehensive details for all experiments presented in Section 5.

### D.2.1. BASE MODEL PRETRAINING

All experiments in Section 5 use a pretrained Qwen2.5-0.5B model trained on a digit-sequence transformation task. Table D1 summarizes the pretraining configuration. To avoid biasing the model to expect operations in specific input positions, we prepend each training sample with a random number of padding tokens, within a limit of 128 tokens per sample.

*Table D1.* Base model pretraining configuration. The model was trained in two phases: Phase 1 on single operations, Phase 2 on compositions of 1–3 operations.

| Parameter | Phase 1 | Phase 2 |
|---|---|---|
| Base Model | Qwen/Qwen2.5-0.5B | |
| PEFT Method | LoRA (r=32, $\alpha$=32) | |
| Target Modules | q, k, v, o, gate, up, down | |
| Training Samples | 100,000 | 500,000 |
| Test Samples | 500 | 500 |
| Operations per Sample | 1 | 1–3 |
| Epochs | 3 | 3 |
| Batch Size | 64 | 64 |
| Learning Rate | 2e-4 | 2e-4 |
| Warmup Steps | 500 | 500 |
| Operations | [ASC], [DESC], [ADD], [SUB], [POL], [REV], [ID] | |
| Sequence Lengths | 2, 3, 4, 6, 8 (held-out: 5, 7, 9) | |
| Held-out 3-op combinations | – | 25% |

### D.2.2. SKILL NEOLOGISMS

**Insertion function**   In Section 5, the insertion function $\phi$ simply swaps the tokens corresponding to the target skill (e.g. "[SHIFT]") with the skill tokens of length $l$ in the prompt.

### D.2.3. COMPOSITIONAL TRANSFER EXPERIMENTS (FIGURE 4)

Table D2 summarizes the configuration for each method in Figure 4.

*Table D2.* Configuration for compositional transfer experiments (Figure 4). All methods learn one of [SHIFT] and [INV-POL] and are evaluated on compositions with held-out pretrain operations.

| Parameter | Skill Neologisms | Prompt Tuning | LoRA |
|---|---|---|---|
| Trainable Structure | Vocab. tokens | Prefix tokens | LoRA adapters |
| Soft Tokens Length/Rank | 20 | 20 | r=16 |
| Trainable Params | 17,920 | 17,920 | $\sim$2.9M |
| Initialization | Mean of pretrain op. embeddings | | – |
| Training Samples | 100,000 | | |
| Validation Samples | 1,000 | | |
| Test Samples | 200 per Sequence Length and permutation | | |
| Operations per Sample | 1–3 (requires $S_{new}$ + 0–2 from $\Sigma_{train}$) | | |
| Held-out Skill | One operation per run (6 total scenarios) | | |
| Sequence Lengths | 2, 3, 4, 6, 8 (held-out: 5, 7, 9) | | |
| Epochs | 3 | 3 | 3 |
| Learning Rate | 5e-3 | 5e-3 | 1e-4 |
| Batch Size | 32 | 32 | 32 |
| Temperature at Inference | 0 (greedy) | | |
| Eval. Metrics | Acc. on $\mathcal{C}_2(S_{new}, \Sigma_{train})$ (ID) | | |
| | Acc. on $\mathcal{C}_2(S_{new}, S_{held\text{-}out})$ (OOD) | | |

**Dataset Configuration:** Training samples compose $S_{new}$ with operations from $\Sigma_{train}$ (6 of the 7 pretrain operations, with one held out). Training and validation data is distributed equally across operation counts (e.g., for max_ops=3, each of 1-op, 2-op, and 3-op receives $\frac{100,000}{3} \approx 33{,}333$ samples).

**Test Dataset Generation:** The test dataset evaluates all permutations of operation orderings to ensure the model learns composable skills rather than memorizing specific sequences. For each $k \in \{2, 3\}$ operations:

- Each sample requires exactly one $S_{new}$, one $S_{held\text{-}out}$, and $(k-2)$ operations from $\Sigma_{train}$

- The order of these 2 (resp. 3) operations is set by sampling one of the 2 (resp. 6) permutations of $S_{new}$, $S_{held\text{-}out}$, and $S \in \Sigma_{train}$.

- $N_{test} = 200$ samples are generated for each sequence length and permutations, yielding 400 test samples per sequence length for k=2 and 1200 test samples per sequence length for k=3.

Each method is trained on 6 configurations (one per held-out operation) for each of the 2 new skills, yielding 12 runs per method.

### D.2.4. MULTI-SKILL COMPOSITION EXPERIMENTS (FIGURE 5)

Table D3 presents the experimental setup for Figure 5.

### D.2.5. ABLATION STUDIES

**Effect of Training Composition Complexity**   Table D4 shows how varying the maximum number of operations during training (max_ops) affects generalization. Other parameters are the same as in Section D.2.3 under the "Skill Neologisms" column.

*Table D3.* Configuration for multi-skill composition experiments (Figure 5). Skill neologisms for `[SHIFT]` and `[INV-POL]` are learned independently, then composed zero-shot.

| Parameter | Value |
|---|---|
| **Skill Neologisms** | |
| Training | Two independently trained skills (config from Table D2) given a held-out skill $S_{\text{held-out}}$, repeated for 6 different $S_{\text{held-out}}$ |
| Composition Method | Insert both skill tokens into test prompts |
| Evaluation | Repeat across all 6 $S_{\text{held-out}}$ |
| **In-Context Learning Baseline** | |
| Examples per Skill | $N \in \{10, 20, 50, 100\}$ |
| Total Examples | $2N$ (N for each target skill) |
| Examples Pool Size | 10,000 samples per skill |
| **Test Dataset** | |
| Test Samples | 50 per sequence length |
| Sequence Lengths | 2–8 |
| Operations per Sample | `[SHIFT]` and `[INV-POL]` |
| Temperature | 1 |

*Table D4.* Effect of training composition complexity. Each row shows results for a different max_ops value during training. All configurations use skill length 20.

| max_ops | Epochs | Training Samples |
|---|---|---|
| 1,2,3 | 2 | 100,000 |

**Evaluation:** Each configuration is evaluated on both 2-operation and 3-operation compositions with held-out skills. The table in the paper reports mean accuracy across all 6 held-out operations for each $S_{new}$.

**Effect of Initialization Method**  Table D5 compares initialization strategies for skill token embeddings. Other parameters are the same as in Section D.2.3 under the "Skill Neologisms" column.

*Table D5.* Initialization method comparison. Both methods use skill length 20, learning rate 5e-3, and 2 epochs of training.

| Method | Description |
|---|---|
| From Pretrain | Mean of pretrain operation embeddings |
| Random | Random Gaussian initialization with $\sigma = 0.2$ |

**Evaluation:** Average accuracy on $\mathcal{C}_2$ and $\mathcal{C}_3$ compositions across all 6 held-out operations.

**Effect of Skill Token Length (Figure 6)**  Figure 6 shows how skill token capacity affects learning and generalization. Parameters are the same as in Section D.2.3 under the "Skill Neologisms" column, while only varying the skill token length $l \in \{1, 5, 10, 20, 50, 100, 200\}$.

### D.2.6. DATASET AND EVALUATION DETAILS

**Operations:** All experiments use 7 pretrained operations on digit sequences:

- `[ASC]`: Sort digits in ascending order

- `[DESC]`: Sort digits in descending order

- `[ADD]`: Add 1 to each digit (mod 10)

- `[SUB]`: Subtract 1 from each digit (mod 10)

- `[POLARITY]`: Map odd digits to 1, even to 0

- `[REVERSE]`: Reverse digit order

*Table D6.* Configuration for length ablation experiments.

| Parameter | Value |
|---|---|
| Skills Evaluated | `[SHIFT]`, `[INV-POL]` |
| Fixed Held-out Skill | `[ADD]` |
| Training Samples | 100,000 |
| max_ops | 2 (1 or 2 operations per sample) |
| Epochs | 1 |
| Learning Rate | 5e-3 |
| Batch Size | 32 |
| Metrics | ID: Acc. on $\mathcal{C}_2(S_{new}, \Sigma_{train})$ |
| | OOD: Acc. on $\mathcal{C}_2(S_{new}, \text{ADD})$ |

- `[ID]`: Identity (no transformation)

Two new operations are learned in all main experiments:

- `[SHIFT]`: Right-shift digits cyclically

- `[INV-POL]`: Map odd digits to 0, even to 1

**Sequence Lengths:**

- Training: 2, 3, 4, 6, 8

- Held-out: 5, 7, 9

**Sample Format:** Each sample follows the pattern `[OP-1]...[OP-k]xxxx=yyyy`, where `xxxx` is the input digit sequence and `yyyy` is the result of applying operations sequentially.

**Evaluation Metrics:**

- **Exact Match Accuracy:** The model must generate the complete correct output sequence.

### D.2.7. COMPUTATIONAL RESOURCES

**Model:** Qwen/Qwen2.5-0.5B

- Embedding Dimension: 896

- Hidden Size: 896

- Layers: 24

- Attention Heads (Q / KV): 14 / 2

- Tie Embeddings: Yes

**Framework:**

- HuggingFace Transformers

- PEFT library for LoRA

- Custom implementation for skill neologisms and prompt tuning (same implementation for both, simply inserting soft tokens before every prompt for prompt tuning)

- Weights & Biases for experiment tracking

**Hardware:** Experiments were run on a NVIDIA RTX 6000 GPU (48GB VRAM).

## D.3. Experimental Details: Section 6

We run the Skill-mix benchmark (Yu et al., 2024) based on the reference implementation[1].

### D.3.1. SKILL-MIX EXPERIMENT: MODUS PONENS AND STATISTICAL SYLLOGISM

We evaluate `meta-llama/Llama-3.2-3B-Instruct` on the public Skill-Mix benchmark using the target skills `modus ponens` and `statistical syllogism`.

**Training set construction.** The training data is constructed as follows. For $k = 2, 3$, we mix each target skill with a subset of skills from the publicly released skills in the Skill-mix repository.

- **Target skills:** `modus ponens` and `statistical syllogism`.

- **Training set size:** 300 GPT-5 generated samples per skill, split evenly across $k = 1, 2, 3$ with 100 samples each.

- **Train skills:** `self serving bias`, `red herring`, `spatial reasoning`, and `folk physics` `(common knowledge physics)`.

**Method hyperparameters.** The method-specific training settings are:

- **Skill Neologisms:** $l = 20$ soft tokens per skill; SFT for 10 epochs; learning rate $5 \times 10^{-3}$; batch size 8; gradient accumulation 2; linear learning rate scheduler; random initialization.

- **Prompt Tuning:** same $l = 20$ tokens prefix length and hyperparameters as skill neologisms.

- **LoRA:** rank 16, LoRA alpha 16, dropout 0.1, targeting `q_proj`, `k_proj`, `v_proj`, `o_proj`, `gate_proj`, `up_proj`, and `down_proj`.

**Testing setup.** We evaluate on 50 test $k = 2$ queries with both target skills jointly, using `temperature=1.0`, and GPT-5 with `temperature=1.0` and `reasoning_effort=minimal` for the grading.

### D.3.2. PROMPTS

We use the same prompts template for inference and grading as the original Skill-mix implementation, reproduced in Listings 1 and 2.

*Listing 1.* Prompt template for Skill-Mix queries.

```
Greetings! I am interested in natural language processing and I was wondering if you
could help me generate an example of text that illustrates multiple skills in semantics
or syntax. The example should be a minimal natural piece of text with up to a few lines
in the context of {topic} that illustrates all of the following skills: {skills_str}.
Please keep the text as short as possible, and make sure the concepts can be found fully
from the text.

For reference, here are the definitions and examples for the concepts:
{skills_definitions_and_examples}

Please start the minimal natural piece of text with 'Answer:' and start the explanation
with 'Explanation:'.

Thanks very much!
```

*Listing 2.* Prompt template for Skill-Mix grading.

```
Greetings! I was wondering if you could help me grade the following answer given by a
student.
```

---

[1] https://github.com/LeoYu/skill-mix/

```
 Ill  first describe the question that was given to the student, and then give you the
students answer, and the grading rubric.

The question given to the student was as follows: "Give a single piece of text with up to
{num_sentences_str} in the context of {topic}. This single piece of text should
illustrate all of the following skills: {skills_str}."

The student's answer was: "{student_answer}"

For reference, here are the definitions for the skills:
{skills_defs_and_examples_simple}

Using a rubric table format, please grade the student's answer with positive scoring.
Each criterion is worth 1 point. The criteria are: {rubric_items}. The table should only
have the following columns: 'Criteria', 'Points Earned'. In the 'Points Earned' column,
please provide only numerical values with no additional formatting. Please introduce the
table with 'Here's the grading table:' and please include a row for 'Total Points Earned'
at the end of the table. Finally, please start your grading explanation with
'Explanation':
```

Below is an example prompt with skills "modus ponens" and "statistical syllogism" on the topic of "gardening":

```
Greetings! I am interested in natural language processing and I was wondering if you
could help me generate an example of text that illustrates multiple skills in semantics
or syntax. The example should be a minimal natural piece of text with up to a few lines
in the context of Gardening that illustrates all of the following skills: modus ponens,
statistical syllogism. Please keep the text as short as possible, and make sure the
concepts can be found fully from the text.

For reference, here are the definitions and examples for the concepts:
**modus ponens**: A syllogism that is of the form "If P then Q. P. Hence Q." For example,
"If today is Tuesday, then John will go to work. Today is Tuesday. Therefore, John will
go to work."
**statistical syllogism**: A syllogism that argues, using inductive reasoning, from a
generalization true for the most part to a particular case. For example, "Almost all
people are taller than 26 inches. Gareth is a person. Therefore, Gareth is taller than 26
inches."

Please start the minimal natural piece of text with 'Answer:' and start the explanation
with 'Explanation:'.

Thanks very much!
```

For skill neologisms, we insert skill tokens based at all the location where the exact skill name appears in the prompt. For example, inserting neologisms of length $l = 20$ for "modus ponens" and "statistical syllogism" in the prompt above results in the following input prompt, where <|skill_id-i|> denotes the i-th soft token for that skill neologism.

```
Greetings! [...] the following skills: <|modus_ponens-0|>...<|modus_ponens-19|>,
<|statistical_syllogism-0|>...<|statistical_syllogism-19|>. Please keep the text as short
as possible, and make sure the concepts can be found fully from the text.

For reference, here are the definitions and examples for the concepts:
**<|modus_ponens-0|>...<|modus_ponens-19|>**: [...]
**<|statistical_syllogism-0|>...<|statistical_syllogism-19|>**: [...]

Please start [...]
```

