# OpenReview forum: "Skill Neologisms: Towards Skill-based Continual Learning"
_ICML.cc/2026/Conference — ICML 2026 spotlight_

### Official Review · Reviewer_SSA4 · 2026-03-05

**Soundness:** 4
**Presentation:** 4
**Significance:** 3
**Originality:** 2
**Overall Recommendation:** 5
**Confidence:** 4

**Summary:**

Authors focus on the topic of continual learning for LLMs: how to extend a given model capability ? Importantly how to do it in a compositional way i.e. how to learn skills that can be combined ? Authors propose to study skill neologisms as a solution, i.e. training composable vocabulary-level soft tokens by leveraging a skill-centric training procedure. Authors show on a controlled operations on digits dataset that skill neologisms outperform LoRa or Prompt Tuning. Skill neologisms can be composed, even if compositions include skills unseen during training or compositions of independently learned skill neologisms.

**Compliance With Llm Reviewing Policy:**

Affirmed.

**Final Justification:**

I will maintain my accept, as the overall rebuttal reinforced this initial assessment: this paper is interesting and ICML-worthy.

**Key Questions For Authors:**

Is my understanding of PT vs Skill Neologisms correct ? If yes, could this very close link between PT and Skill Neologism be stated clearly in your manuscript, early on ?

Related to these questions: what is the precise difference between the PT baseline vs the Skill Neologisms method in figure 4 ?

Do you plan to release your code and learned models ?

**Limitations:**

Authors properly discuss limitations of their work

**Strengths And Weaknesses:**

## Soundness

Experiments are well presented and performed, authors provide enough detail to be able to reproduce their work (although sharing the code would be better). Authors properly introduce and motivate their idea.

## Presentation

The paper reads very well. figures are very helpful. I enjoyed reading it.

One weakness from my point of view is moving the related to the appendix. Crucially, I had trouble understanding the exact difference between classical Prompt Tuning vs Skill Neologism.
From what I understood, PT and Skill Neologism both rely on learning one or several soft tokens. Their are only two differences:
1) Training regime --> Skill Neologism is about training soft-tokens through a skill-centric approach.
2) Insertion in prompt --> While PT are usually prepended, Skill Neologism are included in the vocabulary and as such can be positioned anywhere it is relevant depending on the task.
**Is this understanding correct ? If yes, could this very close link between PT and Skill Neologism be stated clearly in your manuscript, early on ?** I now details are given in section 2.3 but more would be needed.

Minor:
I wonder whether all the discussion and data about the REV operator, which ended up being too complex for the model used, could be moved to the appendix. This would make the introduction of experiments in the main body smoother. It feels like an experimental detail.

l.38: PEFT ? define acronym
l40. "(Arora & Goyal, 2023; Yu et al., 2024; Chen et al., 2023)" --> better to have citations in chronological order
l.72 "In this work, investigate"


## Significance

Given the widespread use of LLMs, finding ways to extend their capabilities in a continual learning manner is a relevant research endeavour. Sharing the code would strengthen the significance of this work. Also, as stated by the authors, further experiments in more natural problem domains would be helpful to assess the generality of their findings.

## Originality

This is were my expertise is limited (I look forward to see other reviewers' perspectives), however to me while the idea is powerful, it can be seen as a relatively simple modification of Prompt Tuning approaches (from what I understood of the difference between the two, cf my previous question). As such, originality is limited, i.e. "fair" level. It does not mean this subject is not worth studying.

---

> ### Author Rebuttal · Authors · 2026-03-31
>
> We thank the reviewer for the positive feedback and the helpful suggestions to improve the manuscript.
>
> # On the difference between Prompt Tuning and Skill Neologisms
> Thank you for pointing this out; we agree that highlighting this distinction early is important. Your understanding is correct: the main technical differences between Skill Neologisms and PT are (1) the focus on skill-centric training and (2) the semantic integration of soft tokens in the prompt, as opposed to the semantically disconnected prefix tokens of PT.
>
> However, we argue that these two differences point to a deeper conceptual distinction. PT is designed to adapt a model to a task; the learned prefix is task-specific and cannot be reused outside its training setting. In contrast, Skill Neologisms aim to learn a reusable, composable representation of a skill. Both technical differences are crucial to achieve this: skill-centric training provides an inductive bias to learn a general representation of the target skill that composes flexibly with other model skills (P2), while semantic integration allows the model to interact with the skill's procedural knowledge via in-context learning, enabling zero-shot combination of independently trained skill tokens (P3), which is something PT cannot achieve by design. Our new SkillMix experiment (see response to Reviewer qyLc, W1) further illustrates this key property of Skill Neologisms in a more complex and realistic natural language setting.
>
> Regarding **Figure 4** specifically: since both methods are trained on the same skill-centered dataset and optimize the same number of soft tokens, the difference in performance is only due to the location of the soft tokens -- prepended as a prefix in PT vs semantically inserted at the location of the target skill in Skill Neologisms.
>
> We will acknowledge the technical similarities of the two methods and clarify this conceptual distinction more clearly in the Introduction and Section 2.3 in the updated manuscript.
>
>
> # On moving some experimental details to the appendix and other minor comments
> Thank you for these suggestions and for noting these inaccuracies. We will move the extended discussion about the REV operator to the appendix to streamline the main text, and will address the remaining inaccuracies in the updated manuscript.
>
>
> # On sharing the code and the learned model
> We are releasing the full codebase for this project to ensure reproducibility: https://anonymous.4open.science/r/skill-neologisms-FC11
> We confirm that we will also share our learned models and skill tokens upon acceptance.

---

> > ### Author Rebuttal · Reviewer_SSA4 · 2026-04-01
> >
> > I thank authors for their response, which addressed all my concerns. I will raise my score to accept.

---

### Official Review · Reviewer_qyLc · 2026-03-12

**Soundness:** 2
**Presentation:** 3
**Significance:** 2
**Originality:** 3
**Overall Recommendation:** 4
**Confidence:** 3

**Summary:**

The key issue examined by this paper is whether new composable skills can be added to a pretrained LLM without weight updates, by learning vocabulary-integrated soft tokens that represent procedural knowledge. The paper first argues that some pretrained tokens already encode procedural knowledge, using XOR/XNOR keyword experiments, and then evaluates the proposed method on a controlled synthetic digit-transformation setting. The main claim is that skill neologisms outperform LoRA and prompt tuning on out-of-distribution composition, and also enable zero-shot composition of independently trained new skills.

**Compliance With Llm Reviewing Policy:**

Affirmed.

**Final Justification:**

thanks for the responses and they have solved my comments. I will modified the score to "weak accept".

**Key Questions For Authors:**

1. The paper’s central claims are validated in a synthetic digit-transformation setting where skills are explicit by construction. Do the authors have any evidence that skill neologisms also work in more realistic language settings where skills are latent and less cleanly separable?

2. The method assumes access to skill-centered datasets. In practice, how sensitive is performance to noise in skill annotation or imperfect filtering of examples for the target skill?

3. The paper argues that pretrained vocabulary tokens can already encode procedural knowledge, based on XOR/XNOR keyword experiments. Could the authors clarify how strong this evidence really is? In particular, how much of the observed gain is due to procedural knowledge stored in the keyword itself versus simpler lexical or frequency effects?

4. LoRA performs poorly on OOD composition in the paper’s setup. To what extent is this a property of LoRA itself versus a consequence of training on task-centric rather than skill-centered data?

5. For Property 3, the paper compares against ICL as a natural zero-shot baseline. Did the authors also consider stronger composition-oriented baselines, such as alternative soft-prompt composition schemes or steering-token style methods?

6. The token-length ablation suggests that limited capacity acts as an inductive bias for composability. Do the authors have a more mechanistic explanation for why this happens? For example, is the degradation at large lengths due to memorization of training compositions, interference with prompt structure, or something else?

**Limitations:**

I think the main paper should state even more explicitly that the current evidence is limited to a synthetic procedural setting, and that broader claims about scalable continual learning in realistic LLM use cases remain preliminary at this stage.

**Strengths And Weaknesses:**

Strengths: The paper asks a clear and interesting question about skill-based continual learning, and the proposed formulation around Properties 1–3 is easy to follow. The controlled experimental design is a strength: the synthetic setting makes the underlying skills explicit, allows exact ID/OOD splits, and cleanly tests both compositional transfer and multi-skill zero-shot composition. Within this controlled regime, the empirical results are fairly convincing. The ablations on token length, composition complexity, and initialization are also useful.

Weaknesses: The strongest results are shown only in a highly controlled synthetic algorithmic domain, so it remains unclear how well the proposed approach transfers to more realistic language tasks where skills are implicit, entangled, and harder to annotate. Besides, an important part of the method is the assumption that one can construct skill-centered datasets in practice; while the paper discusses possible routes, this remains a substantial bottleneck. Lastly, the paper demonstrates that skill neologisms work better than the chosen baselines in this setting, but it does not yet establish that the learned tokens capture reusable “skills” in a deep semantic sense beyond this procedural domain.

---

> ### Author Rebuttal · Authors · 2026-03-31
>
> We thank the reviewer for the thoughtful and constructive feedback.
>
> # On W1 & Q1
>
> To demonstrate the applicability of skill neologism to a realistic language setting, we conducted new experiments on the **SkillMix** [1] benchmark, which specifically evaluates an LLM’s ability to combine natural language skills (e.g., generating a short text that simultaneously incorporates "metaphor" and "statistical syllogism"). We run Llama3.2-3B-Instruct on the public SkillMix repo and select two skills for which the base model has low accuracy: ”modus ponens” and “statistical syllogism”. For each target skills, we construct N=300 training samples by mixing the target skill with 0, 1 or 2 randomly sampled skills (while holding out the other target skill) and collecting high-quality answers using GPT-5. We train LoRA, Prompt Tuning and Skill Neologisms via SFT and test on queries requiring both skills simultaneously.
>
> The results are shown [here](https://ibb.co/v4CpGGHj). Each trained model improves performance on the target skill they were trained for, but not for the other held-out target skill. Only Skill Neologisms successfully allows for the zero-shot combination of both independently learned skill tokens, achieving high accuracy on both skills simultaneously despite never being trained on them jointly. This demonstrates that skill neologisms extend to more realistic language settings.
>
>
> # On W2 and W3: Access to skill-centered datasets and applicability beyond procedural domain
>
> The new experiment demonstrates that skill-centered datasets can be practically constructed in complex natural language settings by using a strong LLM to generate text given a predetermined list of skills (an approach similarly used in [2] for instruction tuning). Therefore, while the need for skill-centered data is an important assumption, we argue that it can be satisfied even in complex, non-algorithmic domains.
>
>
> # On Q2
>
> While precisely evaluating the quality of a skill-centered dataset in the general sense is an open question that we leave for future works, we conducted a new ablation to provide some insights in our controlled setting. We train skill neologisms for $\texttt{INV-POL}$ while randomly replacing occurrences of the target skill in the training set by a random other skill and updating the ground-truth answer accordingly. The results are shown [here]( https://ibb.co/3YNBQNw9), and demonstrate that noise in the skill-centered dataset degrades performance, particularly for OOD composition with new skills.
>
>
>
>
> # On Q3
> The XOR/XNOR experiment from Section 4 compares the “Examples+Keyword” prompt to “Examples” and “Examples+Text Description” prompts. The in-context example and textual description of the XOR operation control for lexical or semantic knowledge effects, and therefore this result demonstrates that the “XOR” tokens contains procedural knowledge **over and above** these simpler effects. While we do not make any broad claims, this simple experiments illustrates that some pretrained tokens can encode procedural knowledge in a wide range of open-source LLMs.
>
> # On Q4
> We would like to clarify that the LoRA model Figure 4 is trained on the same skill-centered dataset as the other methods. As discussed in the paper, we suggest that its poor performance on OOD compositions is due to overfitting the training distribution instead of learning a composable representation of $S_{new}$.
>
> # On Q5
>
> Soft-prompt composition such as SCP [3] learns vocabulary-level soft tokens for zero-shot attribute-object composition in VLMs, but trains all tokens jointly on a shared dataset of compositions. Our P3 is strictly harder: each skill token is trained in a fully independent process with no exposure to other skill tokens. Steering tokens combine in the model's latent activation space. While they can be added together mechanistically, their compositionality relies on activation-space arithmetic rather than the model's inherent in-context abilities. To our knowledge, no existing method attempts zero-shot composition of independently trained soft tokens encoding procedural knowledge in LLMs.
>
>
>
> # On Q6
> While a full mechanistic investigation is beyond the scope of this paper, we hypothesize that excessive capacity (large $l$) allows skill tokens to memorize specific compositions seen during training rather than learning a composable representation of the target skill. Under this view, limited capacity acts as a bottleneck that forces the token to learn only what is invariant across all training compositions (which is precisely the target skill itself), leading to better OOD generalization.
>
>
> [1] Yu et al. "SKILL-MIX: a Flexible and Expandable Family of Evaluations for AI Models." ICLR 2024.
>
> [2] Kaur et al. “Instruct-SkillMix: A Powerful Pipeline for LLM Instruction Tuning” ICLR 2025
>
> [3] Nayak et al. "Learning to Compose Soft Prompts for Compositional Zero-Shot Learning." ICLR 2023

---

> > ### Author Rebuttal · Reviewer_qyLc · 2026-04-03
> >
> > thanks for the responses and they have solved my comments. I will modified the score to "weak accept".

---

### Official Review · Reviewer_Hibq · 2026-03-13

**Soundness:** 3
**Presentation:** 3
**Significance:** 3
**Originality:** 4
**Overall Recommendation:** 5
**Confidence:** 3

**Summary:**

This paper introduces "skill neologisms," a method for skill-based continual learning in LLMs. The core idea is to learn new, composable skills by optimizing soft tokens (neologisms) integrated into the model's vocabulary, while keeping the base model weights frozen. The method is motivated by the observation that pre-trained LLMs already possess tokens associated with procedural knowledge (e.g., "XOR"). The authors propose that training these soft tokens on "skill-centered" datasets—where every sample requires the target skill mixed with other mastered skills—can yield skill representations that are composable with out-of-distribution (OOD) skills (Property 2) and can be combined zero-shot with other independently learned skill neologisms (Property 3), all without weight updates (Property 1). Experiments on a controlled algorithmic task demonstrate that skill neologisms outperform baselines like LoRA and Prompt Tuning in OOD compositional transfer and enable zero-shot multi-skill composition, validating the proposed properties.

**Compliance With Llm Reviewing Policy:**

Affirmed.

**Key Questions For Authors:**

1. Generalization to Natural Language: What are the most immediate challenges in applying skill neologisms to a natural language task (e.g., a specific style of reasoning, a complex formatting skill)? Could you outline a concrete, feasible next-step experiment beyond synthetic tasks?
2. Skill Interaction and Interference: In your multi-skill composition experiment, skills are applied sequentially. How would the method handle skills that need to be interleaved or applied in a more interdependent manner? Could negative interference arise when composing many skill tokens in a single context?
3. Defining and Scoping a "Skill": The success of the method hinges on defining a learnable "skill." In your view, what are the boundaries? Could a very broad "skill" (e.g., "chain-of-thought reasoning") be learned this way, or is the method better suited for more granular, procedural operations?
4. Comparison to Alternative Parameter-Efficient Methods: Did you consider or experiment with other PEFT methods like (IA)^3 or Adapter Layers in your baseline comparisons? Their modular nature might also allow for some form of composition.

**Limitations:**

The authors are generally upfront about limitations, which I largely echo: the synthetic experimental setting, the dependency on skill-centered datasets, the computational cost of training soft tokens, and the inherent instability/sensitivity of optimizing soft prompts. A further limitation is the lack of exploration into how the order or positioning of multiple skill tokens in the prompt affects compositional performance.

**Strengths And Weaknesses:**

Strengths:
1. Novel and Well-Motivated Concept: The idea of extending a model's vocabulary with learnable skill tokens is creative and builds nicely on prior work (neologisms, skill composition theory). The initial evidence that pre-trained tokens like "XOR" already function as skill tokens is compelling.
2. Clear Problem Formulation: The paper clearly defines the three desired properties (P1-P3) for skill-based continual learning, providing a concrete framework for evaluation.
3. Rigorous and Controlled Experiments: The synthetic digit-operation task is well-suited for a proof-of-concept. It allows for clean ID/OOD splits and unambiguous evaluation of compositionality. The experiments are thorough, comparing against strong baselines (LoRA, Prompt Tuning, ICL) and including informative ablations (token length, training composition complexity).
4. Strong Empirical Results: The key results are convincing: skill neologisms show significantly better OOD generalization than LoRA/Prompt Tuning and enable zero-shot composition of independently learned skills, outperforming in-context learning.

Weaknesses:
1. Limited Scope and Realism: The evaluation is confined to a synthetic, algorithmic domain with explicitly defined skills. It remains unclear how well the method would scale to natural language tasks where skills are implicit, noisy, and intertwined.
2. Dataset Construction Hurdle: The requirement for "skill-centered" datasets is a significant practical limitation. While the paper suggests ways to construct them (using LLM metacognition, existing labels), this adds complexity and may not be feasible for many arbitrary skills.
3. Computational Cost Acknowledged but Not Quantified: The paper notes that training soft tokens requires full backpropagation, making it computationally comparable to fine-tuning for large models. However, no concrete runtime or cost comparisons with baselines are provided.
4. Ablation on Initialization is Light: The initialization ablation shows random initialization still works, but the difference for INV-POL is notable. More analysis on why certain skills might be more sensitive to initialization would be helpful.

---

> ### Author Rebuttal · Authors · 2026-03-31
>
> We thank the reviewer for the positive feedback and insightful comments. Please see our responses to your questions below.
>
> # On Q1 & Q2: Generalization to natural language; more complex skill compositions
>
> To address these important next steps, we have conducted a new experiment using the SkillMix [1] environment, targeting two complex natural language skills ("modus ponens" and "statistical syllogism") with a Llama3.2-3B-Instruct model. We train one skill neologism per target skill independently on N=300 skill-centered samples (holding out the other target skill), then compose them zero-shot at test time to test P3 in a more complex and realistic setting. Full details and results are provided in our response to Reviewer qyLC (W1).
>
> Since the SkillMix task requires generating a single coherent text that simultaneously incorporates multiple skills, the skills are inherently interleaved and interdependent. While LoRA and Prompt Tuning each improve on their respective target skill but fail to compose both simultaneously, Skill Neologisms successfully allow the model to compose these independently learned skills zero-shot. These results confirm that Skill Neologisms extend beyond controlled algorithmic settings: interleaved, interdependent natural language skills can be learned independently and composed zero-shot.
>
> # On Q3: defining and scoping a “skill”
>
> This is an important question. While we do not offer a definitive answer at this stage, we note that the literature suggests the skill-centered view operates effectively across different levels of granularity: from specific mathematical manipulations [2] to higher-level capabilities such as instruction-following skills [3] and reasoning-related skills [4]. Whether Skill Neologisms can be learned equally effectively across all granularity levels is an interesting open question that we leave for future work.
>
>
> # On Q4: alternative PEFT methods
> Thank you for these interesting suggestions. A key distinction between Skill Neologisms and methods like IA^3 and Adapter Layers is that they operate on the model's internal computations through activation rescaling or weight updates, whereas Skill Neologisms compose *contextually* via the model's in-context learning abilities. As a result, independently trained skill neologisms can be combined at inference time simply by placing multiple tokens in the prompt, with no joint training or architectural intervention required. This contextual composition mechanism is precisely what enables P3, and is not naturally achievable by methods that operate within the model's weight space.
>
>
> [1] Yu et al. "SKILL-MIX: a Flexible and Expandable Family of Evaluations for AI Models." ICLR 2024
>
> [2] Didolkar et al. "Metacognitive capabilities of llms: An exploration in mathematical problem solving." NeurIPS 2024
>
> [3] Kaur et al. “Instruct-SkillMix: A Powerful Pipeline for LLM Instruction Tuning” ICLR 2025
>
> [4] Gandhi et al. "Cognitive Behaviors that Enable Self-Improving Reasoners, or, Four Habits of Highly Effective STaRs." COLM 2025

---

> > ### Author Rebuttal · Reviewer_Hibq · 2026-04-04
> >
> > Thank authors for their response, which addressed all my concerns. I keep my score as accept.

---

### Decision · Program_Chairs · 2026-04-30

**Decision:**

Accept (spotlight)

**Comment:**

The authors presents a elegant and novel framework for compositional continual learning in LLMs.

The reviewers initially expressed concerns regarding the limited scope of the experiments (synthetic tasks only), the authors' rebuttal convincingly addressed these concerns by providing new, high-quality results on the SkillMix benchmark. This new evidence confirms the method's utility in interleaved, interdependent natural language settings. Additionally, the authors successfully argued the distinct value proposition of their method over existing PEFT methods like Prompt Tuning by highlighting the modularity and zero-shot compositionality of their approach. The reviewers' consensus is unanimous in favor of acceptance.